# Characteristics of CD133-Sustained Chemoresistant Cancer Stem-Like Cells in Human Ovarian Carcinoma

**DOI:** 10.3390/ijms21186467

**Published:** 2020-09-04

**Authors:** Chao Lien Liu, Ying Jen Chen, Ming Huei Fan, Yi Jen Liao, Tsui Lien Mao

**Affiliations:** 1School of Medical Laboratory Science and Biotechnology, College of Medical Science and Technology, Taipei Medical University, Taipei 11031, Taiwan; chaolien@tmu.edu.tw (C.L.L.); m609104006@tmu.edu.tw (M.H.F.); yjliao@tmu.edu.tw (Y.J.L.); 2PhD Program in Medical Biotechnology, College of Medical Science and Technology, Taipei Medical University, Taipei 11031, Taiwan; 3Division of General Internal Medicine and Geriatrics, Department of Internal Medicine, Chang Gung Memorial Hospital, Taoyuan 33305, Taiwan; ma2875@cgmh.org.tw; 4School of Medicine, College of Medicine, Chang Gung University, Taoyuan 33305, Taiwan; 5Department of Pathology, College of Medicine, National Taiwan University, Taipei 10002, Taiwan

**Keywords:** ovarian cancer (OC), cancer stem cell (CSC), CD133, chemoresistance, sphere-forming assay

## Abstract

Cancer stem cells (CSCs) are considered to be the origin of ovarian cancer (OC) development, recurrence, and chemoresistance. We investigated changes in expression levels of the CSC biomarker, cluster of differentiation 133 (CD133), from primary OC cell lines to induction of CSC-spheres in an attempt to explore the mechanisms related to modulation of stemness, drug resistance, and tumorigenesis in CSCs, thus facilitating the search for new therapeutics for OC. The effect of CD133 overexpression on the induction of CSC properties was evaluated by sphere-forming assays, RT-qPCR, flow cytometry, cell viability assays, and in vivo xenograft experiments. Moreover, the potential signaling molecules that participate in CD133 maintenance of stemness were screened by RNA-sequencing. CD133 expression was upregulated during OCSC induction and chemotherapeutic drug treatment over time, which increased the expressions of stemness-related markers (SOX2, OCT4, and Nanog). CD133 overexpression also promoted tumorigenesis in NOD/SCID mice. Several signalings were controlled by CD133 spheres, including extracellular matrix receptor interactions, chemokine signaling, and Wnt signaling, all of which promote cell survival and cell cycle progression. Our findings suggest that CD133 possesses the ability to maintain functional stemness and tumorigenesis of OCSCs by promoting cell survival signaling and may serve as a potential target for stem cell-targeted therapy of OC.

## 1. Introduction

Ovarian carcinoma (OC) is the most lethal gynecologic malignancy that affects a large population of women worldwide [1,2]. Around 70–90% of patients with advanced OC relapse within two years [1], and recurrent tumors become chemoresistant, underscoring the need for new treatment options. Recurrence is believed to be caused by the presence of residual tumor-propagating cells that cannot be completely eradicated by surgical and/or pharmacological regimens [3]. Accumulating evidence suggests that some of these residual cancer cells have stem cell (SC)-like properties, such as self-renewal and differentiation [4,5], and are responsible for disease recurrence after the first-line treatment [6].

In OC, cancer SCs (CSCs) are reported to be important in tumor initiation, dissemination, and recurrence, and targeting CSC components is an endeavor of OC therapy [7,8,9]. Identifying CSCs relies on the presence of SC markers, and in OC, many biomarkers can be used to confirm the presence of CSCs. However, various studies have been carried out on different types of OC with heterogeneous cell populations, such as tumor cell lines, primary tumors, tumor xenografts, and in vitro tumor spheres and with various methodologies. Biomarkers include cluster of differentiation 144 (CD144) with or without CD117 [4,10], Hoechst-excluding cells (the “side population”) [11], CD24 [12], epithelial cell adhesion molecule (EpCAM) [10], prominin-1 (CD133) [13,14,15], and aldehyde dehydrogenase-1A1 (ALDH1A1) [16], the expressions of which are either enriched or inconsistent in different types of OC-CSC cells, thereby indicating a dynamic process in which different populations of cells are involved in forming a cellular hierarchy [17]. Moreover, it is known that CSC phenotypes are not uniform amongst the various cancer types and even in tumors of the same histological type, and they can change via in vitro culture conditions [18]. Cell heterogeneity within tumors may influence the disease course and the response to treatment (known as drug resistance) [19]. There are many other questions regarding CSCs that should be further investigated: specifically, how does the tumor microenvironment influence CSC marker expression, and how is cancer stemness maintained.

In this study, CD133 expressing sphere (SP-CD133) cells maintained a significant chemoresistance ability. The effects of CD133 in promoting sphere formation and tumor growth were observed in both in vitro and in vivo experiments. Furthermore, we also elucidated the underlying molecular mechanisms of SP-CD133 cells in promoting sphere formation and tumorigenesis using RNA-sequencing (RNA-Seq) with real-time reverse-transcription quantitative polymerase chain reaction (RT-qPCR) confirmation. Results suggested that SP-CD133 reduction of extracellular matrix (ECM) receptor interactions/focal adhesion in combination with induction of chemokine signaling/Wnt signaling molecules facilitates tumor growth by promoting cell survival and/or cell cycle progression. These findings provide a potential therapeutic strategy for treating drug-resistant OCs.

## 2. Results

### 2.1. CD133 Expression Was Associated with the Sphere-Forming Capacity and Induced Stemness in Human OC Cells

The effects of sphere-forming culture on the stemness of four human OC cell lines (SKOV3, HTB75, OVCAR3, and IGROV1) were first investigated. As shown in Figure 1A, compared to the formation of the monolayer phenotype under normal culture conditions on day 0, the sphere-forming culture clearly resulted in the formation of ovarian spheres in SKOV3 and IGROV1 cells by day 16. CD133 protein expression levels in both SKOV3 and IGROV1 cells were significantly positively associated with the number of spheres formed (** *p* < 0.01, * *p* < 0.05, respectively; Figure 1B,D), while comparisons of CD44 levels in all cell lines were insignificant during the sphere-forming culture period. Moreover, expressions of stemness markers in the four cell lines and sphere cells formed were examined using RT-qPCR analysis. Our results showed that mRNA expression levels of core pluripotency factors Sox2, Oct4, and Nanog in both CSC-like SKOV3 and IGROV1 cells were significantly higher than those in the CSC-like HTB75 and OVCAR3 cells (Figure 1C; * *p* < 0.05, ** *p* < 0.01, *** *p* < 0.001). In addition, mRNA expression levels of stemness markers in the four CSC-like cell lines were also significantly higher than those in monolayer phenotype cells, indicating that monolayer phenotype cells almost did not express stemness markers.

Similarly, we determined the cell surface CSC markers of CD44 and CD133 using flow cytometric analysis, and results showed a significant positive correlation between CD133 protein expression and sphere numbers in the SKOV3 cell line at different induction time points (Appendix A). Taken together, these findings showed that the generation of SKOV3 and IGROV1 spheres significantly induced expressions of the CD133 CSC marker and Sox2, Oct4, and Nanog stemness-related markers.

### 2.2. Pretreatment with Cisplatin (CDDP)- and/or Paclitaxel (PTX)-Enhanced CD133 Expression and Drug Resistance in CSC-Like SKOV3 Spheres

CSCs are highly associated with chemoresistance to anticancer drugs [20], and tumors collected from platinum-resistant patients exhibit higher CD133 expression [21]. Thus, we investigated whether pretreatment of SKOV3 cells with CDDP and/or PTX promoted the sphere-forming ability and resistance against CDDP/PTX on CSC-like SKOV3 spheres. As shown in Figure 2A,B, SKOV3 cells pretreated with 1 μM CDDP alone, 1 nM PTX alone, or the combination for different time periods exhibited enhanced expression of CD133 in CSC-like SKOV3 spheres compared to the untreated or isotype controls (* *p* < 0.05; ** *p* < 0.01; *** *p* < 0.001, Figure 2B and Appendix A), suggesting that these chemicals promoted the sphere-forming ability. In addition, the combination of CDDP and PTX synergistically promoted higher expression of CD133 in CSC-like SKOV3 spheres than either single drug alone (Figure 2B). Moreover, CD133-expressing CSC-like SKOV3 spheres (CD44^+^/CD133^+^ spheres) were further resistant to CDDP alone, PTX alone, and the combination of both drugs compared to the SKOV3 parental (SKOV3-P) control and SKOV3 spheres that did not express CD133 (CD44^+^/CD133^−^ spheres) following incubation for 12, 24, 36, and 48 h (* *p* < 0.05; ** *p* < 0.01; *** *p* < 0.001, respectively, Figure 2C–F). These data suggest that both CD133 expression and sphere formation are essential for maintaining CSC traits and chemoresistance capacities. Likewise, a similar effect of CD133 was also observed in another OC cell line, IGROV1, where the CSC-like IGROV1 spheres (CD44+/CD133+) were significantly better in maintaining CSC traits and chemoresistance abilities than CD44+/CD133^−^ IGROV1 spheres (Appendix A).

### 2.3. CD133 Promotes CSC Traits and Is Essential for the Chemoresistant Capacity

Because pretreatment with CDDP/PTX enhanced CD133 expression and promoted the sphere-forming ability and drug resistance in CSC-like SKOV3 spheres, the essential roles of CD133 in drug resistance and CSC traits in both parental SKOV3 (SKOV3-P) cells and CSC-like SKOV3 spheres were further explored by transduction of CD133 (SKOV3-CD133^+^, Appendix A). As shown in Figure 3A, serial sphere formation assays revealed that CD133 overexpression greatly enhanced morphologic changes (Figure 3B) and cell numbers (Figure 3C). In drug-resistant cell viability assays, CD133 overexpression in CSC-like SKOV3 spheres (CD44^+^/CD133^+^ spheres) were more resistant to 1.2 μM CDDP alone, 1.4 nM PTX alone, and their combination compared to CSC-like SKOV3 spheres without CD133 expression (CD44^+^/CD133^−^ spheres) and SKOV3-P controls following 12, 24, 36, and 48 h of drug treatment (* *p* < 0.05; ** *p* < 0.01; *** *p* < 0.001, respectively, Figure 3D–G). Additionally, CD133 overexpression in SKOV3-P (SKOV3-CD133^+^) cells also induced higher CDDP/PTX resistance compared to non-transduced SKOV3-P cells (* *p* < 0.05; ** *p* < 0.01, respectively, Figure 3D–G). Collectively, these data again confirmed that CD133 and sphere formation are both essential for maintaining stemness-associated phenotypes and drug resistance.

### 2.4. Overexpression of CD133 Enhanced the In Vivo Tumor Growth of CSC-Like SKOV3 Spheres

Based on the observation that CD133 improves the sphere-forming ability and drug resistance in OC cell lines, the in vivo effect of CD133 on the growth of tumors derived from CSC-like OC spheres was further investigated. To test the efficacy of tumor growth, we divided experimental NOD/SCID mice into three groups (*n* = 6 per group), and they were subcutaneously (s.c.) injected with SKOV3 spheres with CD133 transduction (SP-CD133), SKOV3 spheres with mock transduction (SP-Mock), or parental SKOV3 (SKOV3-P) cells. The experimental schedule is shown in Figure 4A. A significantly enhanced tumor growth effect was observed in mice injected with SP-CD133 spheres compared to mice treated with SP-Mock and SKOV3-P groups over time (*** *p* < 0.001, Figure 4B). We also observed increased tumor growth in mice implanted with SP-Mock compared to mice implanted with SKOV3-P (* *p* < 0.05, Figure 4B). In addition, there were no significant difference in body weights among the three mice groups during the experimental period (Figure 4C). At week 9, tumor volumes and tumor weights had significantly increased in both sphere (CSC-like) cell-implanted groups, especially the SP-CD133-implanted mouse group, compared to control mice implanted with SKOV3-P cells (*n* = 6 per group; *** *p* < 0.001 and * *p* < 0.05, respectively, Figure 4D,E). Similar to our in vitro study, the in vivo study suggested that CSC-like spheres per se have the potential to increase tumor growth compared to non-sphere parental cells. In addition, CSC-like spheres with CD133 overexpression further dramatically increased in vivo tumor growth.

Since CSC-like SKOV3 spheres transduced with CD133 expression (SP-CD133) significantly promoted tumor growth, we also detected expressions of stemness-related markers in mice xenografts harvested at the study endpoint. Results revealed that the CSC markers of CD44 and CD133 and stemness-related markers of Nanog and Oct4 were significantly overexpressed in tumors derived from SP-CD133 implanted mice (* *p* < 0.05 and ** *p* < 0.01, Figure 5) compared to tumors derived from SP-Mock- and SKOV3-P-implanted controls. Moreover, we also examined the Ki67 proliferation index, which also showed significantly higher values in tumors from SP-CD133 implanted mice (* *p* < 0.05, Figure 5).

### 2.5. Reductions in ECM Receptor Interactions/Focal Adhesion in Combination with Induction of Cell Survival/Cell Cycle Progression Caused by CD133 Overexpression

To identify genes modulated by CD133 overexpression in CSC-like SKOV3 spheres, RNA-Seq was conducted. Results showed that compared to parental SKOV3 (SKOV3-P) and SKOV3 spheres with mock transduction (SP-Mock) groups, 109 genes were upregulated, and more than 480 genes were downregulated in the CD133-transduced SKOV3 sphere (SP-SKOV3) group (Figure 6A). The EKGG pathway enrichment analysis of differentially expressed genes (DEGs) revealed that ECM receptor interactions, focal adhesion, phosphatidylinositol 3-kinase (PI3K)/AKT signaling pathway, chemokine signaling pathway, and Wnt signaling accounted for the majority of DEGs (Figure 6B). To further investigate the candidate mechanisms related to SP-CD133-induced tumorigenesis, DEGs belonging to pathways including ECM receptor interaction, focal adhesion, and cancer signaling-related molecules from Figure 6B were extracted and are presented in a signaling diagram (Figure 6C). Subsequently, a real-time RT-qPCR analysis was used to verify signaling relationships among the DEGs mentioned above (Figure 6D). Results revealed that CSC-like SKOV3 with CD133 overexpression produced downregulation of ECM receptor interaction/focal adhesion molecules, including integrin-alpha6 (ITGA6), integrin-beta8 (ITGB8), PI3KCA, and forkhead box O3 (FOXO3). Meanwhile, compared to SKOV3-P and SP-Mock controls, other important factors involving the chemokine signaling pathway and/or Wnt signaling were significantly induced, which promote cell cycle progression and cell survival, including FasL, RBL2, G-protein-coupled receptor (GPCR), cAMP response element-binding protein (CREB), Bcl2, lipoprotein-related receptor protein5/6 (LRP5/6), and transcription factor (TCF)/lymphoid enhancer-binding factor (LEF) (Figure 6D). These results indicated that reductions in focal adhesion molecules together with induction of cell survival/cell cycle progression factors significantly promoted SP-CD133 xenograft tumor growth. As expected, sphere cells pre-treated with CDDP/PTX, especially those treated with their combination, showed similar mRNA expression profiles as SP-CD133 cells following RT-qPCR confirmation (Figure 6D).

## 3. Discussion

OC is the second most common gynecologic malignancy but has the highest mortality due to presentation in late stages in most patients [2]. Despite modest improvements in response rates following certain conventional therapies, the overall survival for patients with advanced OC remains disappointing [22]. It is believed that a rare population of cells, namely CSCs, are responsible for chemoresistance and tumor recurrence [23]. Moreover, elimination of this CSC population is thought to be a beneficial strategy to increase therapeutic responses. Therefore, researchers have been devoted to investigating the molecular mechanisms underlying the CSC-related chemoresistance of OC. So far, accumulating evidence shows that CD133 (the most common ovarian CSC marker)-expressing CSCs are involved in the development of OCs [13,14,24,25,26]. However, a complete understanding of CD133-expressing CSCs in regulating OC development and chemoresistance remains controversial.

One previous study directly showed that primary OC specimens were composed of low densities of cells expressing CSC markers such as ALDH1A1, CD44, and CD133, whereas tumors collected immediately after primary therapy were more densely populated with these markers [21]. Moreover, CD133 expression was higher in tumors collected from platinum-resistant patients [21]. These findings support the standpoint that in OC, CSC subpopulations contribute to tumor chemoresistance, thereby ultimately leading to recurrent disease. A series of studies attempted to identify membrane markers that could characterize OC initiating cells. However, most of those studies had major limitations caused by the use of very heterogeneous cells, including cultured primary tumor cells [4,27], multiple passage primary xenografts [28], or immortalized cell lines [11,16]. Among the various markers tested, Kryczek et al. explored putative SC markers, including CD133, CD117, CD44, ABCG2, epithelial-specific antigen (ESA), and ALDH, in primary OC samples and found that (i) these markers exhibited highly heterogeneous expressions, and (ii) only CD133 and ALDH were enriched in cancer-initiating cells, as verified both in vitro by a tumor sphere assay and in vivo inoculation into NOD/SCID mice [29]. Indeed, our results showed that CD133 expression was essential and significantly correlated with sphere induction, which was also responsible for chemoresistance and tumorigenicity capabilities. Interestingly, although CD44 is not an essential SC marker along with sphere induction, we found that the two sphere-inducible cell lines, SKOV3 and IGROV1, were inherently loaded with high levels of CD44 compared to the other two non-inducible cell lines, OVCAR3 and HTB75 (Figure 1B). As More and Alvero reported, OC tissues exhibited heterogeneous CD44 expression [30]. They found a large-sized and non-dividing CD44^+^ ALDH1^+^MYD88^+^ subtype of OCs, which exhibits several properties of CSCs, including the capacity to generate tumors in immunodeficient mice, the capacity of functioning as tumor vascular progenitors, and chemoresistance [30]. Nevertheless, further investigations are needed due to the plasticity of OC SCs and the complexity of defining this cell population.

Standard first-line chemotherapy for OC consists of the combined administration of cisplatin (CDDP) and paclitaxel (PTX). CDDP is a DNA strand cross-linking drug that generates DNA damage and consequent activation of cyclin inhibitors such as p21, with cell cycle arrest at the G1 or G2 phase, whereas PTX is a mitotic inhibitor, which promotes the formation/stabilization of microtubules, with consequent cell cycle blockage at metaphase or anaphase. Moreover, mechanisms of resistance to CDDP were studied, including increased glutathione and metallothionein levels, increased drug efflux, decreased drug uptake, increased DNA repair, and tolerance to the formation of platinum-induced DNA adducts [31], whereas PTX treatment can induce chemoresistance and acquisition of the epithelial-to-mesenchymal transition (EMT) and a CSC-like phenotype in OC [32,33]. On the other hand, experimental studies showed that short-term single exposure to standard chemotherapy treatment (CDDP, PTX, or both agents) induced chemotherapy-surviving cells, which were more capable than untreated cells of forming tumors in immunodeficient mice, and those cells maintained this property over time [34]. Those results are consistent with our findings that CSC-like SKOV3 spheres pretreated with CDDP and/or PTX were enriched in CD133 and favored chemoresistance in vitro, which also endowed CD133^+^CSC-like spheres with a high tumorigenic potential in vivo.

One of the main features of OC CSCs, like CSCs of other organs, consists of increased expression of transcription factors essential for induction of the stemness phenotype. Among these, NANOG expression is reported to be involved in OC progression through regulation of E-cadherin and FOXJ1, resulting in reduced chemosensitivity and poor survival [35]. Interestingly, activation of FOXO3 nuclear localization and consequent inhibition of the expressions of stemness markers, including Nanog, Oct4, and c-MYC, are caused by inhibition of the formation of ovarian tumor spheroids [36]. On the other hand, an important role of SOX2, a transcription factor, is in maintaining the CSC state of OC cells; furthermore, SOX2-expressing OCs display enhanced resistance to apoptosis induced by chemotherapy or tumor necrosis factor (TNF)-related apoptosis-inducing ligand (TRAIL)-induced cell death [37]. Our findings confirmed that levels of Nanog, Oct4, and SOX2 were increased in CD133-CSC spheres, whereas FOXO3 expression was reduced in the CD133-expressing CSC-like spheres, thereby facilitating their capabilities of stemness maintenance, self-renewal, and chemoresistance. Elucidation of the molecular regulations of OCs would help define the development of precursor lesions and allow us to understand the mechanisms underlying the effects of CD133 transduction and sphere formation, ultimately leading to chemoresistance and/or tumorigenicity. The development of high-throughput next-generation sequencing (NGS) technology offered the opportunity to characterize these potential mechanisms. As numerous pathways were identified by the KEGG analysis, we focused on ones that may be involved in stemness regulation and tumorigenesis in OCs and are potentially targetable. Among those targets, ITGA6, ITGB8, and PI3KCA, which function in ECM receptor interactions, and focal adhesion molecules showed significant reductions in CD133-SP cells, whereas GPCR/CREB, a chemokine signaling pathway, and LRP5/TCF, the Wnt signaling pathway, were significantly increased in SP-CD133 cells, demonstrating the regulation of stemness and tumorigenesis via cell survival as well as cell-cycle progression. Integrins are a superfamily of cell adhesion receptors that bind to ECM ligands, and integrin-mediated signaling regulates proliferation, survival, and invasion of tumorigenic cells with unclear mechanisms [38,39]. In OC, Zhu et al. showed that ITGA6 and ITGB5 expressions were significantly downregulated in patients with high-grade serous OC (HGSOC) compared to those in normal counterparts [39]. In contrast, Zhang et al. reported that signaling cascades connecting oncogenic K-Ras with ITGA6 functions modulated cancer cell survival and tumorigenesis [40]. Moreover, ITGB8 was reported to be associated with drug resistance, by which overexpression of ITGB8 restored CDDP resistance inhibited by miR-199a-3p in SKOV3 cells [41], thereby reducing overall survival of OC patients [42]. Further work is needed to explore the complex and conflicting functions of different integrin subunits in HGSOC. Furthermore, with regards to OC, Wnt signaling is involved in normal and tumor development of ovarian and fallopian tube SCs, and is reported to promote both stemness and chemoresistance [43,44].

OC cells contain a subpopulation of stem-like cells with the ability to grow in an anchorage-independent manner in vitro and are able to form tumors in vivo. The existence of highly tumorigenic and chemotherapy-resistant CSC-regulated biomarkers opens a path to targeting these cells, leading to minimization of drug resistance and tumor relapse. Therefore, considering personalized medicine in the treatment of OC, potential CSC targets could serve as new therapeutics, including but not limited to SC markers, SC signaling pathways needed for renewal and/or survival, and the SC niche. Advanced studies investigating the contributions of CSC subpopulations and signaling pathways to maintaining CSC traits will direct future therapeutic target designs.

## 4. Materials and Methods

### 4.1. Cell Lines

Human OC cell lines (SKOV3, OVCAR3, HTB75, and IGROV1) were purchased from American Type Culture Collection (ATCC, Manassas, VA, USA) and maintained in Dulbecco’s modified Eagle’s medium (DMEM; Life Technologies, Carlsbad, CA, USA) supplemented with 10% fetal bovine serum (FBS; HyClone, Logan, UT, USA) at 37 °C with 5% CO_2_ incubation. All cell lines were tested to identify the species and detect mycoplasma, and authentication was confirmed using a short-tandem repeat (STR) profile analysis by ATCC. According to a histotype analysis of OC cell lines by genetic and immunohistochemical study, IGROV1 was classified as an endometrioid type and was found to have mutations of TP53, PTEN, and ARID1A. SKOV3, bearing mutations of PTEN and ARID1A, was classified as an atypical non-serous/clear/endometrioid type. Both OVCAR3 and HTB75 have mutation of TP53 and belong to high-grade serous carcinoma [45].

### 4.2. Sphere-Formation Assay

For the sphere-formation assay, cells were harvested and resuspended in a defined serum-free medium (DMEM/F12) (Corning, Corning, NY, USA) containing 0.3% bovine serum albumin (BSA; Sigma-Aldrich, St. Louis, MO, USA), 5 μg/mL insulin (Thermo Fisher Scientific, Carlsbad, CA, USA), 10 ng/mL epidermal growth factor (EGF) (Thermo Fisher Scientific), 10 ng/mL basic fibroblast growth factor (bFGF) (Thermo Fisher Scientific), and 12 ng/mL leukemia inhibitory factor (LIF) (Thermo Fisher Scientific), and plated in triplicates into 1% chitosan (Sigma-Aldrich)-precoated 6-well plates (Corning) at a density of 2 × 10^5^ cells/well. Every 3 days, growth factors were replenished with fresh culture medium. Sphere cells were counted under an Olympus BX50 light microscope (Tokyo, Japan) at each indicated time points.

### 4.3. Flow Cytometry

Cancer cell lines and cancer SC (CSC)-like cells were collected and prepared according to the manufacturer’s instructions. The following monoclonal antibodies were obtained from BD Biosciences (Los Angeles, CA, USA): phycoerythrin (PE) mouse anti-human CD44, and allophycocyanine (APC) mouse anti-human CD133, and their immunoglobulin G (IgG) controls. Reacted cells were analyzed using an Attune^TM^ NxT Acoustic Focusing Cytometer (Thermo Fisher Scientific), and each flow cytometer measurement used AttuneTM NxT software (Thermo Fisher Scientific) for data processing. Individual CD44+CD133^−^ sphere and CD44+CD133+ sphere populations were sorted using a FACSAria III cell sorter (BD Biosciences, San Jose, CA, USA) from CD44+ spheres obtained from the sphere-forming assay. Three independent experiments were performed for the statistical analysis.

### 4.4. Cell Viability Assay

Cell viability was determined by a 3-(4,5-dimethylthiazol-2-yl)-2,5-diphenyltetrazolium bromide (MTT; Thermo Fisher Scientific) assay as previously described [46]. Briefly, cells were seeded into 96-well plates, cultured with complete medium for 16 h, and then treated with certain concentrations of cisplatin (CDDP) (1.0 or 1.2 μM; Sigma-Aldrich), paclitaxel (PTX) (1.0 or 1.4 nM; Sigma-Aldrich), or their combination for a series of time points. Results of the 50% inhibitory concentration (IC50) titration of CDDP/PTX for SKOV3 and SKOV3-CD133 cells are shown in Appendix A. After removing the supernatant, DMEM containing 5 mg/mL MTT was added and incubated for 4 h. The resulting formazan was dissolved in 1 mL isopropanol, and the absorbance at 570 nm was measured using a microplate reader (Bio-Rad Laboratories, Hercules, CA, USA). The number of viable cells was proportional to the absorbance, and cell viability was presented as a percentage of the dimethyl sulfoxide (DMSO) control (untreated).

### 4.5. In Vitro Overexpression of CD133

The CD133 (GenBank: NM_006017) gene was purchased from the National RNAi Core Facility (Taipei, Taiwan) and used in construction of the pLAS5w.Pbsd recombinant lentivirus-expressing vector (Invitrogen, Waltham, MA, USA). SKOV3 cells at a density of 10^5^ cells/well were seeded in 24-well plates. After overnight incubation, 2 mL of fresh media containing 8 μg/mL polybrene (Sigma-Aldrich) and the recombinant lentivirus were added. Following transduction and puromycin (Sigma-Aldrich) selection, SKOV3 cells with stable CD133 overexpression were used for subsequent assays.

### 4.6. Assessment of Tumor Growth Using a Xenograft Model

Six ~8-week-old non-obese diabetic/severe combined immunodeficiency (NOD/SCID) (NOD.CB17-prkdc^scid^/JNarl) mice were maintained in a specific pathogen-free (SPF) facility according to National Institute of Health guidelines for animal care and guidelines of National Taiwan University (Taipei, Taiwan). All animal experiments were performed according to protocols reviewed and approved by the Institutional Animal Care and Use Committee (IACUC/IACUP; protocol no: 20150527). To establish xenograft models, mice were subcutaneously inoculated with 10^7^ of attached SKOV3 cells, mock-transduced SKOV3-sphere cells, and/or CD133-overexpressing SKOV3-sphere cells on the left flank on day 0. Tumor dimensions were measured using calipers, and the tumor volume (V) was calculated with the formula *V* = 1/2 (length × width^2^). Xenografts were fixed with formalin and embedded in paraffin, and sections were cut and processed for immunohistochemical (IHC) staining.

### 4.7. IHC and Immunoreactivity Scoring

Solid tumors were excised from sacrificed tumor xenograft mice under sterile conditions at indicated times following xenograft implantation. For IHC, serial tissue sections (4 μm thick) were prepared from formalin-fixed xenograft SKOV3-P, SP-Mock, and SP-CD133 tumor samples and mounted on glass slides. After rehydration, sample sections were stained with monoclonal antibodies against human Ki67, CD44, CD133, Nanog, and Oct4. After 10 min of 3,3′-diaminobenzidine incubation, tumors were counterstained with hematoxylin. Images were acquired with a BX50 Olympus microscope. For semiquantitative analysis of the immunoreactivity of markers, an H-score [47] was used in this study. Briefly, more than 10 fields were counted in each case, and the H-score was subsequently counted by multiplying the percentage and intensity of stained cells [strongly stained (3×), moderately stained (2×), and weekly stained (1×)], with a possible range of 0–300.

### 4.8. RNA-Seq and Differential Expression Analysis

SKOV3 attached (SKOV3-P) and sphere cells were collected as previously described. CD133^–^sphere cells (SP-Mock) and CD133^+^ sphere cells (SP-CD133) were directly sorted in RNA lysis buffer using a BD FACSAria III cell sorter (BD Biosciences, San Jose, CA, USA). RNA was extracted using a LabPrep^TM^ RNA Plus Mini Kit (Taigen Bioscience, Taipei, Taiwan), and libraries were prepared at the BioTools Microbiome Research Center Next Generation Sequencing facility (BioTools, Taipei, Taiwan) and sequenced on an Illumina HiSeq 4000 as 50-bp single-end reads. Genes that were significantly and differentially expressed were selected based on a multiple of change of >2.0 and a *p* value of <0.05, and subsequently analyzed by a Kyoto Encyclopedia of Genes and Genomes (KEGG) pathway enrichment analysis and protein–protein interaction (PPI) network analysis.

### 4.9. Real-Time Reverse-Transcription Quantitative Polymerase Chain Reaction (RT-qPCR)

Cells were collected and lysed for total RNA extraction using a LabPrep^TM^ RNA Plus Mini Kit (Taigen Bioscience). Complementary DNA (cDNA) was synthesized by reverse-transcription of RNA using High-Capacity cDNA Reverse Transcription Kits (Thermo Fisher Scientific) according to the manufacture’s standard protocols. Briefly, the qPCR was performed in triplicate for each cDNA sample on an Applied Biosystems StepOne^TM^ and StepOnePlus^TM^ Real-Time PCR System (Applied Biosystems; Thermo Fisher Scientific) using PowerUp^TM^ SYBR^TM^ Green Master Mix (Thermo Fisher Scientific). The crossing threshold (Ct) value of the transcripts assessed by the RT-qPCR was normalized to the human *S26* gene. Changes in messenger (m)RNA expression levels are expressed as multiples of change relative to the control ± standard deviation (SD). Primer sequences used are listed in Table 1.

### 4.10. Statistical Analysis

Data were expressed as the means ± SD of three independent experiments. A two-tailed Student’s *t*-test was used for intergroup comparisons. Comparisons between groups were determined with a one-way analysis of variance (ANOVA). Differences were considered significant at *p* < 0.05 using SPSS 13.0 software.

## Figures and Tables

**Figure 1 ijms-21-06467-f001:**
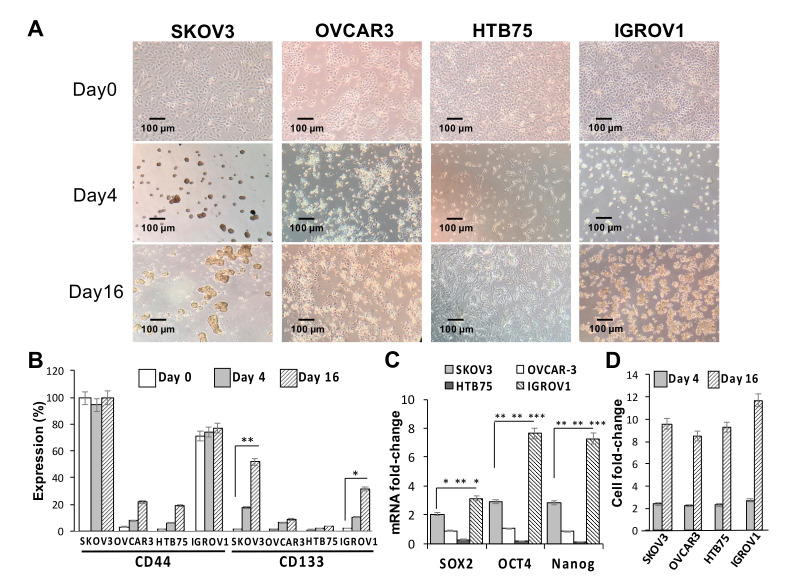
Sphere-forming ability and expression of stemness-related markers of human ovarian cancer (OC) cell lines. (**A**) Optical microscopic images showed morphological changes and sphere formation of SKOV3, OVCAR3, HTB75, and IGROV1 cells at days 0, 4, and 16. Individual scale bars are shown. OC-sphere cells collected at each time point for (**B**) cell surface expressions of cluster of differentiation 44 (CD44) and CD133 were assessed using flow cytometry; (**C**) expression of the stemness-related markers Sox2, Oct4, and Nanog on day 16 were assessed using RT-qPCR; and (**D**) sphere cell numbers. * *p* < 0.05, ** *p* < 0.01, *** *p* < 0.001.

**Figure 2 ijms-21-06467-f002:**
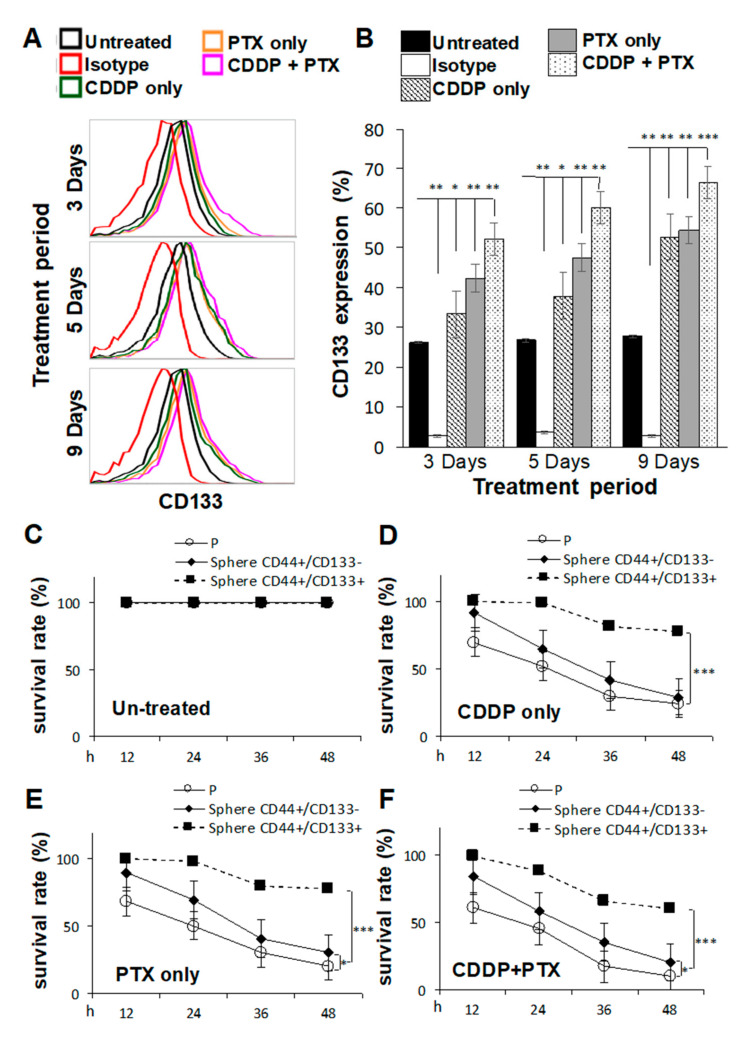
Pretreatment with cisplatin (CDDP) combined with paclitaxel (PTX) synergistically enhanced the sphere-forming ability, cluster of differentiation 133 (CD133) expression, and chemoresistant capacities of cancer stem cell (CSC)-like SKOV3 spheres. SKOV3 parental cells (SKOV3-P) were pretreated with 1 μM CDDP alone, 1 nM PTX alone, or their combination for 3, 5, and 9 days, and then cells were collected for 4-day sphere-forming and cell surface CD133 expression assays in (**A**,**B**). CD44+CD133^−^ sphere as well as CD44+CD133+ sphere populations were sorted for cell viability assay comparisons among Untreated (**C**), CDDP-treated (**D**), PTX-treated (**E**), and CDDP/PTX-treated (**F**) after 12, 24, 36, and 48 h of drug treatment. Presented data were acquired from three independent experiments. * *p* < 0.05, ** *p* < 0.01, *** *p* < 0.001.

**Figure 3 ijms-21-06467-f003:**
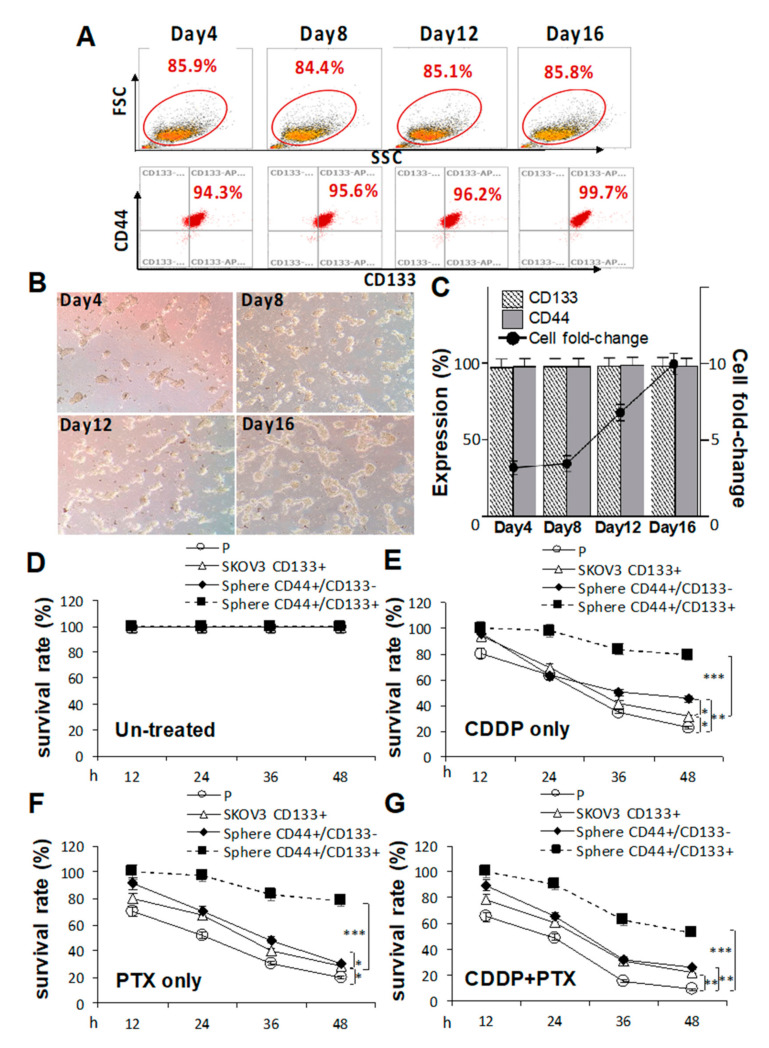
Cluster of differentiation 133 (CD133) expression is essential for sustaining cancer stem cell (CSC) traits and drug-resistance capacities. Effects of CD133 transduction and overexpression in (**A**) sphere-forming ability in (**C**) and morphological changes in CD133 non-expressing SKOV3 cells following 4, 8, 12, and 16 days of sphere cell induction in (**B**). Effects of CD133 overexpression with 1.2 μM cisplatin (CDDP) alone, 1.4 nM paclitaxel (PTX) alone, or their combination were evaluated by cell viability assays among untreated in (**D**), CDDP-treated in (**E**), PTX-treated in (**F**), and CDDP+PTX-treated in (**G**) after 12, 24, 36, and 48 h of drug treatment. Data shown were acquired from three independent experiments. * *p* < 0.05; ** *p* < 0.01; *** *p* < 0.001.

**Figure 4 ijms-21-06467-f004:**
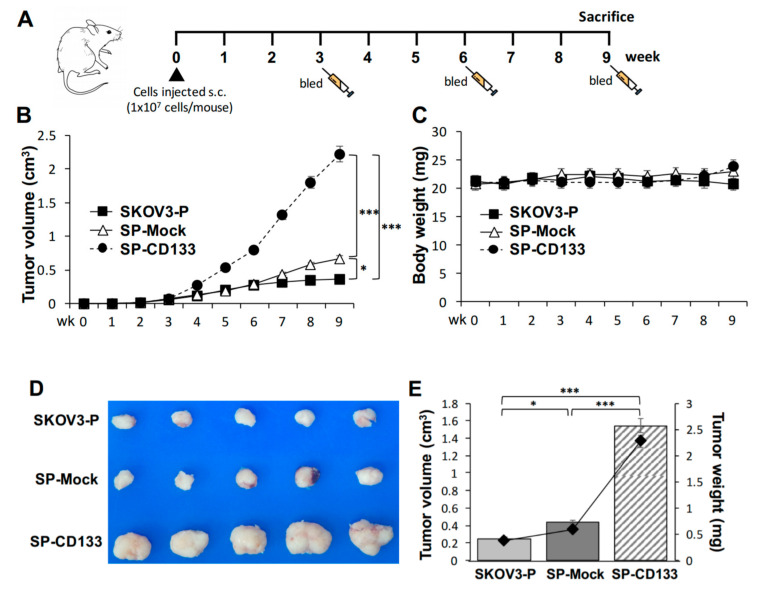
Cancer stem cell (CSC)-like SKOV3 spheres transduced with cluster of differentiation 133 (CD133) promoted tumor growth in a xenograft mice model. SKOV3 spheres with CD133 transduction (SP-CD133), SKOV3 spheres with mock-transduction (SP-Mock), and parental SKOV3 (SKOV3-P) cells were subcutaneously (s.c.) implanted into NOD/SCID mice. (**A**) Detailed treatment schedule of the in vivo study is shown. (**B**) Tumor volumes (cm^3^) during the subsequent period (*n* = 6 per group; * *p* < 0.05; *** *p* < 0.001). (**C**) Body weights during the subsequent period. (**D**) Gross images of dissected tumors at the study endpoint. (**E**) Mean tumor volumes (cm^3^) plotted in a bar graph, and mean tumor weights (mg) expressed in line plots from each mouse group at the study endpoint (*n* = 6 per group; * *p* < 0.05; *** *p* < 0.001).

**Figure 5 ijms-21-06467-f005:**
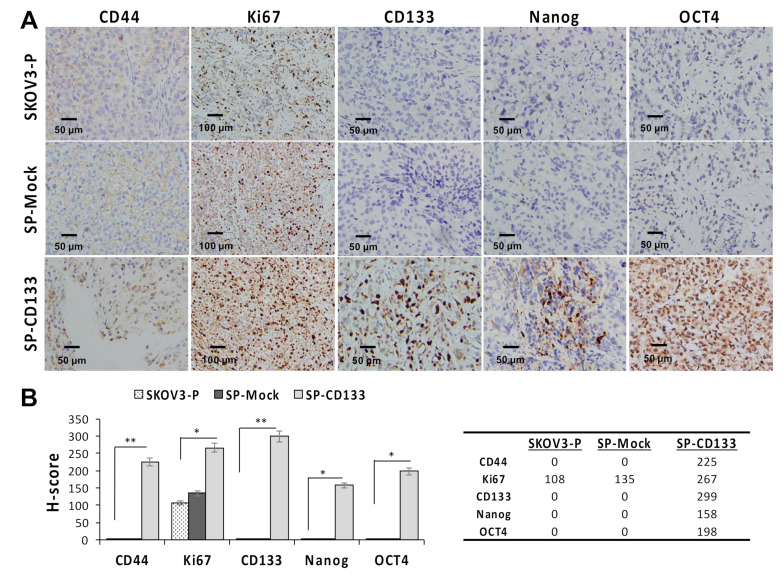
Cell proliferation and stemness related markers are overexpressed in SKOV3 spheres with cluster of differentiation 133 (CD133) transduction (SP-CD133) tumors. Tumors were harvested from mice bearing SP-CD133, SKOV3 spheres with mock-transduction (SP-Mock), and parental SKOV3 (SKOV3-P) subcutaneous (s.c.) xenografts. (**A**) Formalin-fixed, paraffin-embedded tumor sections were consecutively cut and stained for human CD44, Ki67, CD133, Nanog, and Oct4. Images were taken using a camera (DP22) and microscope (BX50) under 400× or 200× magnification. Individual scale bars are shown. (**B**) An average H-scores for each marker and comparison between groups are shown (* *p* < 0.05; ** *p* < 0.01).

**Figure 6 ijms-21-06467-f006:**
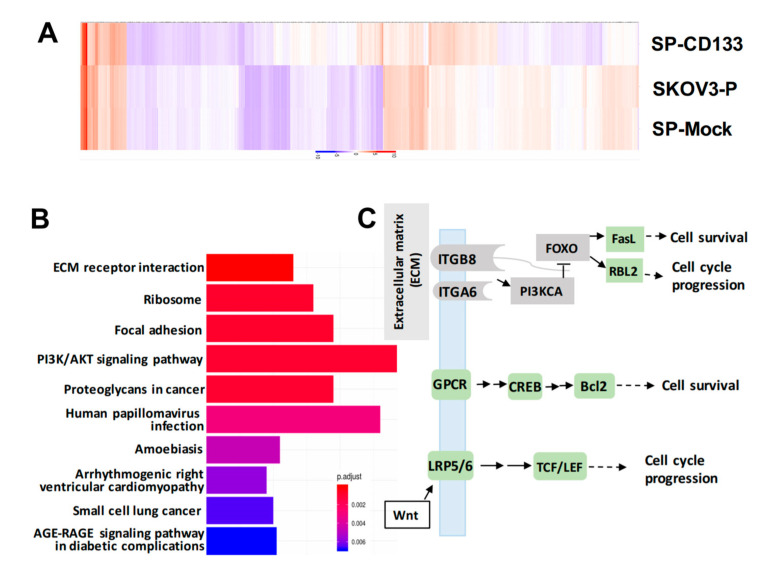
Overexpression of cluster of differentiation 133 (CD133) in SKOV3 spheres with CD133 transduction (SP-CD133) induced the reduction of extracellular matrix (ECM) receptor interactions/focal adhesion and promoted cell cycle progression. (**A**) The heatmap of the differentially expressed genes (DEGs) extracted from the three SKOV3-related modules including SP-CD133, SKOV3 spheres with mock-transduction (SP-Mock), and parental SKOV3 (SKOV3-P) cells. (**B**) KEGG pathway enrichment analysis of all DEGs. (**C**) Simplified diagram of key factors/pathways modulated by CD133. (**D**) Representative RT-qPCR analysis comparing expression levels of ITGA6, ITGB8, PIK3CA, FOXO3, RBL2, FasL, Bcl2, GPCR, CREB, LRP5, and TCF. Data shown were acquired from three independent experiments. * *p* < 0.05; ** *p* < 0.01; *** *p* < 0.001, ns: not significant.

**Table 1 ijms-21-06467-t001:** The primers used for RT-qPCR.

Gene Name	Primer Sequences	Gene Bank No.
CD44-F CD44-R	5′-GCTGACCTCTGCAAGGCTT-3′ 5′-GGATGTACACCCCTGTGTT-3′	NM_000610.4
Sox2-F Sox2-R	5′-CGAGTGGAAACTTTTGTCGGA-3′ 5’-TGTGCAGCGCTCGCAG-3′	NM_003106
Oct4-F Oct4-R	5′-GTGGAGAGCAACTCCGATG-3′ 5′-TGCTCCAGCTTCTCCTTCTC-3′	NM_002701
Nanog-F Nanog-R	5′-ATTCAGGACAGCCCTGATTCTTC-3′ 5′-TTTTTGCGACACTCTTCTCTGC-3′	NM_024865
ITGA6-F ITGA6-R	5′-CCATGCACGCGGATCGAGTT-3′ 5′-GGGATTCCTGCTTCGTATTA-3′	NM_001079818.3
TIGB8-F TIGB8-R	5′-GGACGCTGCCGACTTGTCTT-3′ 5′-GGAAACCACCCTAATGTACA-3′	NM_002214.3
PIK3CA-F PIK3CA-R	5′-CGACCATCATCAGGTGAA-3′ 5′-GCTTCTTGAGTAACACTT-3′	NM_006218.4
FOXO3-F FOXO3-R	5′-GGAGCTGGACCCGGAGTT-3′ 5′-CGTCCTCGTCGTCTTCAT-3′	NM_201559.3
RBL2-F RBL2-R	5′-ATGAGCGAAAGCTACACG-3′ 5′-TGCTCTGAACATTTCAGG-3′	NM_005611.4
FasL-F FasL-R	5′-AAGAGAGATCCAGCTTGC-3′ 5′-TAGATCTGGGGATATGGG-3′	NM_001302746.1
Bcl2-F Bcl2-R GPCR-F GPCR-R	5′-TACTAATAATAACGTGCC-3′ 5′-ATCCAGCTATTTTATTGG-3′ 5′–AATTGGAGGTTTGTTTGC-3′ 5′–ATACCCCAGTTTGACTCC-3′	NM_000633.3
		NM_148963.4
CREB-F CREB-R	5′–AACAAATGACAGTTCAAGC-3′ 5′–GAGACTGAATAACTGATGG-3′	NM_134442.5
LRP5-F LRP5-R	5′–AAGAAGCTGTACTGGACG-3′ 5′–AATCCGGGGCGTCTCACC-3′	NM_001291902.2
TCF-F TCF-R	5′–TTGTGACCAGTATTAAGG-3′ 5′–AAGCCCTGGCCCCAGTCC-3′	NM_001287182.2
S26-F S26-R	5′-CCGTGCCTCCAAGATGACAAAG-3′ 5′-GTTCGGTCCTTGCGGGCTTCAC-3′	NC_016448.1

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
