# Peer review of "Characteristics of CD133-Sustained Chemoresistant Cancer Stem-Like Cells in Human Ovarian Carcinoma"

_ijms, 2020, doi:10.3390/ijms21186467_

Round 1

Reviewer 1 Report

Liu and colleagues aimed to assess the role of CD133 in human ovarian carcinoma stem-like cells. The authors overexpressed CD133 in sphere-forming SKOV3 cells and observed increased potentials for chemotherapy resistance and initiation of tumor growth driven by CD133-overexpression. The authors further applied RNA-Seq to investigate underlying molecular mechanisms and showed extracellular matrix receptor interactions, chemokine signaling and Wnt signaling to be regulated by CD133-overexpression. Despite these interesting findings, the following major issues have to be addressed prior to considering the manuscript for publication:

Major comments

1.) While four different cell lines were applied in the beginning of the study (Fig. 1), the authors utilize only SKOV3 cells throughout the rest of the manuscript. The major findings should be repeated with at least one additional cell line or primary ovarian carcinoma cells, since the present observations may otherwise only be characteristic for SKOV3 cells and not ovarian carcinoma stem-like cells in general.

2.) Figure 1C: While Oct4 and Nanog seem upregulated in SKOV3 and IGROV1 cell lines cultivated as spheres, expression levels of Sox2 are only elevated in SP-SKOV3 cells and not in SP-IGROV1 cells, as claimed by the authors. This discrepancy should be clarified by the authors, particularly since the differential upregulation of Sox2 questions a common mechanism driven by CD133 (see also comment 1).

3.) Figure 2A: The flow cytometry measurements depicted are not convincing. The authors should also provide CD44-measurements, since CD44+ cells are shown in the survival assays (panels C-F). Were CD44+CD133+ and CD44+CD133- sorted from the parental population? If so, respective data should be provided.

Likewise, the authors should indicate which gates were applied exactly in the histograms presented in panel A, since the amount of CD133-positive cells shown in the quantification (panel B) is only hard to observe in the histograms (panel A). In addition, it is unclear to the referee why a small peak left to the major measurement peaks is observable in panels A directly next to the Y-axis. Please show the complete gating strategy.

4.) As a functional assay, the authors overexpressed CD133 in SP-SKOV3 cells already expressing CD133 at high level (figures 1 and 2). Given these high expression levels of CD133 in SP-SKOV3, how do the authors explain the only minor potential for initiation of tumor growth (figure 4B)? In addition, the term “CD44+CD133+” is misleading here, since the term was already used for SP-SKOV3-expressing cells without genetic editing (figure 2). Most importantly, CD133 should be knocked down in SP-SKOV3 cells expressing (not overexpressing) CD133 followed by respective in vitro analysis to prove the hypothesis of CD133 being essential for chemoresistance (and tumor initiation).

Minor comments

1.) The position of the description of columns (“SKOV3-P, Sphere- CD44+/CD133+…”) of the panels C-F in figure 2 and D-G in figure 3 is completely misleading and should be located next to the respective panels.

Author Response

Specific Comments of Reviewer #1

Comments and Suggestions for Authors

Liu and colleagues aimed to assess the role of CD133 in human ovarian carcinoma stem-like cells. The authors overexpressed CD133 in sphere-forming SKOV3 cells and observed increased potentials for chemotherapy resistance and initiation of tumor growth driven by CD133-overexpression. The authors further applied RNA-Seq to investigate underlying molecular mechanisms and showed extracellular matrix receptor interactions, chemokine signaling and Wnt signaling to be regulated by CD133-overexpression. Despite these interesting findings, the following major issues have to be addressed prior to considering the manuscript for publication:

Major comments

1.)While four different cell lines were applied in the beginning of the study (Fig. 1), the authors utilize only SKOV3 cells throughout the rest of the manuscript. The major findings should be repeated with at least one additional cell line or primary ovarian carcinoma cells, since the present observations may otherwise only be characteristic for SKOV3 cells and not ovarian carcinoma stem-like cells in general.

Response: we have included another OC cell line (IGROV1) besides SKOV3 for the investigation of the significance of CD133 in sphere induction and drug resistance. As presented in revised figure 1, sphere induction of IGROV1 significantly increased the expression of CD133 (1B) and stemness-related markers SOX2, OCT4 and Nanog (1C). For functional assays of IGROV1-spheres, IGROV1 cells pretreated with 1uM CDDP alone, 1 nM PTX alone, or CDDP/PTX combination for different time periods (3 and 5 days) exhibited significantly enhanced expression of CD133 by CSC-like IGROV1 spheres compared to the untreated or isotype controls (*P < 0.05, Figure S3A), indicating that these drugs promoted the sphere-forming ability of IGROV1. In addition, the combination of CDDP and PTX synergistically promoted higher expression of CD133 in CSC-like IGROV1 spheres than either single drug alone. Moreover, CD133-expressing CSC-like IGROV1 spheres (CD44+/CD133+ spheres) were more resistant to CDDP alone, PTX alone, and the combination of both drugs compared to the IGROV1 parental (IGROV1-P) control and IGROV1 spheres that did not express CD133 (CD44+/CD133- spheres) following incubation for 24 and 48 h (*P < 0.01, Figure S3B~E). These results suggest a similar effect of CD133 for sphere formation, maintaining CSC traits and chemoresistance capacities in both SKOV3 and IGROV1 cells (revised figure S3, and revised manuscript P 4, Results section 2.2).

2.)Figure 1C: While Oct4 and Nanog seem upregulated in SKOV3 and IGROV1 cell lines cultivated as spheres, expression levels of Sox2 are only elevated in SP-SKOV3 cells and not in SP-IGROV1 cells, as claimed by the authors. This discrepancy should be clarified by the authors, particularly since the differential upregulation of Sox2 questions a common mechanism driven by CD133 (see also comment 1).

Response: We have rechecked our row data of figure 1C, and found that an error occurred during analysis of the relative mRNA fold-change of Sox2 by using SP-SKOV3 as the baseline for comparison (original figure 1C). Actually, the relative expression level of stemness-related markers should be compared to its own parental attached cells, and take respective attached parental cells as the baseline for comparison. We have also re-examined the original calculation data of OCT4 and Nanog. Both used the right baseline control. After correcting the relative fold change of Sox2, we re-plotted figure 1C. All three markers (Sox2, Oct4, and Nanog) were significantly increased with the spheres-induction in both SKOV3 and IGROV1 cells (revised figure 1C). Thank you very much for this helpful comment.

3.) Figure 2A: The flow cytometry measurements depicted are not convincing. The authors should also provide CD44-measurements, since CD44+ cells are shown in the survival assays (panels C-F). Were CD44+CD133+ and CD44+CD133sorted from the parental population? If so, respective data should be provided.

Likewise, the authors should indicate which gates were applied exactly in the histograms presented in panel A, since the amount of CD133-positive cells shown in the quantification (panel B) is only hard to observe in the histograms (panel A). In addition, it is unclear to the referee why a small peak left to the major measurement peaks is observable in panels A directly next to the Y-axis. Please show the complete gating strategy.

Response: Based on our multiple times of experimental results, we found that the expression of CD44 in both attached and sphere SKOV3 cells were more than 97% and had no significant change after sphere induction up to 16 days. Due to space limitation, we put the CD44/CD133 measurement and gating strategy in the supplemental figure 1 (figure S1). To make it clearer to the reviewer, we attach the flow gating strategy during sphere induction using dot-plotting (the original gating) from one of our multiple repeated experiments below.

Again, due to space limitation and to simplify the presented data of figure 2A and B, we only showed CD133 levels at different drug-pretreated condition at each time point (days 3, 5, and 9) using histogram in figure 2A and summarized in figure 2B. Moreover, the histograms-plotted in figure 2A was exactly adopted from the dot-plotted results analyzed using the same gating strategy. Based on the comment of the reviewer, we also included the dot-plot results in revised supplemental figure 2 (revised Figure S2).

For the survival assays in figure 2C~F, different sphere populations (CD44+CD133- spheres and CD44+CD133+ spheres) were isolated using a flow sorter. The sorting strategy and results from a sphere-forming assay at day 10 is also attached in the figure below, where P4 is the CD44+ spheres, P5 (CD44+CD133+ spheres) and P6 (CD44+CD133- spheres) were gated from the P4 population for cell sorting respectively (also described in the revised manuscript P 13, materials and methods section 4.3. Flow cytometry).

4.)As a functional assay, the authors overexpressed CD133 in SP-SKOV3 cells already expressing CD133 at high level (figures 1 and 2). Given these high expression levels of CD133 in SP-SKOV3, how do the authors explain the only minor potential for initiation of tumor growth (figure 4B)? In addition, the term “CD44+CD133+” is misleading here, since the term was already used for SP-SKOV3-expressing cells without genetic editing (figure 2). Most importantly, CD133 should be knocked down in SP-SKOV3 cells expressing (not overexpressing) CD133 followed by respective in vitro analysis to prove the hypothesis of CD133 being essential for chemoresistance (and tumor initiation).

Response: First, we have to clarify two issues:

(1). SKOV3 parental cells express very low level of CD133 (please see figure 1B), thus CD133 knockdown is not feasible in SKOV3 parental cells;

(2). CD133 expression was increased only after sphere induction. CD133 level was increased to around 20% at day 4 of sphere induction, and even to only around 55% at day 16 of sphere induction (please see figure 1B and figure S1).

Although we got CD44+CD133+ SKOV3-spheres (without genetic editing) using a flow sorter, however, another concern is the stability of spheres, specifically in the in vivo experiments. Therefore, we overexpressed CD133 (by genetic editing) in

“SKOV3 parental cells”, and induced spheres by sphere-forming assays, thereby making sure that both SKOV3-parental and -sphere cells can stably express almost 100% CD133 even in the in vivo models. Moreover, CD133 knockdown is difficult to perform due to the dynamic status or/and maintenance condition of spheres.

As for the essential role of CD133 in chemoresistance in vitro, figures 2 (without genetic editing) and 3 (with genetic editing) showed two conclusions: (1). “Sphere” is important for chemoresistance, and CD133 is essential in maintaining functional spheres; (2).“CD133” is essential for chemoresistance, CD133 in both attached- and sphere- cells were significantly more chemoresistant as demonstrated by cell survival assays. For the essential role of CD133 in tumor initiation in vivo, as shown in figure 4, SKOV3-spheres with CD133 expression (SP-CD133) had dramatically increased tumor growth in mice compared to the control mice bearing SKOV3-spheres without CD133 (SP-Mock), thereby illustrating an essential role of CD133 in tumor initiation.

Minor comments

1.) The position of the description of columns (“SKOV3-P, Sphere- CD44+/CD133+…”) of the panels C-F in figure 2 and D-G in figure 3 is completely misleading and should be located next to the respective panels.

Response: In order to make figure 2 and 3 more clear, we re-plotted panels C-F in Fig. 2 and panels D-G in Fig. 3 using the line graphs, as well as the figures in the revised manuscript (revised figure 2 and revised manuscript P 5, legend of figure 2; revised figure 3 and revised manuscript P 6, legend of figure 3).

Reviewer 2 Report

The authors investigated expression levels of CD133 as a CSC marker from ovarian cancer cell lines to induction of CSC-spheres. They showed CD133 increased the expressions of stem cell markers and promoted tumorigenesis in the NOD/SCID mice. And they point out that CD133 possesses the ability to maintain functional stemness and tumorigenesis of ovarian cancer stem cells.

The authors suggested that CD133 overexpression promote cell survival and cell cycle progression and may serve as a potential target for stem cell-targeted therapy of ovarian cancer. This article is interesting for gynecologists and pathologists.

Broad comments:

  1. The manuscript should be arranged as Title page, Abstract, Keywords, Materials and Methods, Results, and Discussion.
  2. Each histotype (serous, mucinous, endometrioid, et al.) of ovarian cancer shows different molecular mechanisms of carcinogenesis. In this study, how did the authors to select four human OC cell lines. And SKOV3 cells derive from which hystotypes.

Specific comments:

    1. P.14, line 385: “formalin-fixed tumor samples” were taken from what tumor tissues? Detailed description is needed.

Author Response

Specific Comments of Reviewer #2

Comments and Suggestions for Authors

The authors investigated expression levels of CD133 as a CSC marker from ovarian cancer cell lines to induction of CSC-spheres. They showed CD133 increased the expressions of stem cell markers and promoted tumorigenesis in the NOD/SCID mice. And they point out that CD133 possesses the ability to maintain functional stemness and tumorigenesis of ovarian cancer stem cells.

The authors suggested that CD133 overexpression promote cell survival and cell cycle progression and may serve as a potential target for stem cell-targeted therapy of ovarian cancer. This article is interesting for gynecologists and pathologists.

Broad comments:

  1. The manuscript should be arranged as Title page, Abstract, Keywords, Materials and Methods, Results, and Discussion.

Response: The sequence of the manuscript was prepared according to the standard format of IJMS which is Title page, Abstract, Keywords, Introduction, Results, Discussion, Materials and Methods, and References.

  1. Each histotype (serous, mucinous, endometrioid, et al.) of ovarian cancer shows different molecular mechanisms of carcinogenesis. In this study, how did the authors to select four human OC cell lines. And SKOV3 cells derive from which hystotypes.

Response: The four OC cell lines used in this study are IGROV1, SKOV3, HTB75 (also known as Caov3) and OVCAR3. We selected these four cell lines due to their similarity in growth rate and accessibility. IGROV1 and SKOV3 were used for further study because of their ability to form spheres whereas HTB75 and OVCAR3 did not form spheres well following sphere-forming assays. According to a histotype analysis of OC cell lines by genetic and immunohistochemical study, IGROV1 was classified as endometrioid type and was found to have mutations of TP53, PTEN, and ARID1A. SKOV3, bearing mutation of PTEN and ARID1A, was classified as atypical non-serous/clear/endometrioid type. Both OVCAR3 and HTB75 have mutation of TP53 and belong to high-grade serous carcinoma (Ref: PLOS ONE, 2013.8(9): e72162). We had added this information in the manuscript (section 4.1. Cell lines)

Specific comments:

  1. 14, line 385: “formalin-fixed tumor samples” were taken from what tumor tissues? Detailed description is needed.

Response: The detailed description of the sentence on P.14 was revised as: formalin-fixed xenograft SKOV3-P, SP-Mock, and SP-CD133 tumor samples”.

Round 2

Reviewer 2 Report

 I’m sorry for misunderstanding. The sequence of the manuscript was prepared according to the standard format of IJMS.

 The manuscript has been significantly improved.